# Adolescent-Reported Latino Fathers’ Food Parenting Practices and Family Meal Frequency Are Associated with Better Adolescent Dietary Intake

**DOI:** 10.3390/ijerph18158226

**Published:** 2021-08-03

**Authors:** Aysegul Baltaci, Silvia Alvarez de Davila, Alejandro Omar Reyes Peralta, Melissa N. Laska, Nicole Larson, Ghaffar Ali Hurtado, Marla Reicks

**Affiliations:** 1Department of Food Science and Nutrition, University of Minnesota, Minneapolis, MN 55410, USA; mreicks@umn.edu; 2Center for Family Development, University of Minnesota Extension, Minneapolis, MN 55411, USA; salvarez@umn.edu (S.A.d.D.); reyes067@umn.edu (A.O.R.P.); 3School of Public Health, University of Minnesota, Minneapolis, MN 55454, USA; nels5024@umn.edu (M.N.L.); larsonn@umn.edu (N.L.); 4School of Public Health, University of Maryland, College Park, MD 20742, USA; ahurtado@umd.edu

**Keywords:** Latino fathers, early adolescents’ consumption, fruit and vegetables, sweets/salty snacks, sugar-sweetened beverages, fast food, fathers’ food parenting practices, family meals

## Abstract

Most studies of food-related parenting practices, parental meal involvement, and adolescent dietary intake have focused on maternal influences; studies of paternal influences, particularly among marginalized groups, are lacking. This study examined lower-income, Latino fathers’ food parenting practices and involvement in planning meals, buying/preparing foods, and family meal frequency, separately and in combination, to identify relationships with adolescent food intake. Baseline data were used from Latino adolescents (10–14 years, n = 191, 49% boys) participating with their fathers in a community-based overweight/obesity prevention intervention. Fathers reported sociodemographic characteristics. Adolescents reported frequency of fathers’ food parenting practices, fathers’ food/meal involvement, and family meals and participated in 24 h dietary recalls. The analysis included regression models using GLM (generalized linear mixed model) and PLM (post GLM processing) procedures. Most fathers were married, employed full-time, and had annual incomes below USD 50,000. Favorable fathers’ food parenting practices were associated with adolescent intake of more fruit and vegetables and fewer sugar-sweetened beverages, sweets/salty snacks, and less fast food (*p* < 0.05 or *p* < 0.01). No independent effects of family meal frequency or fathers’ food/meal involvement were observed on adolescent dietary outcomes. Additional analyses showed favorable food parenting practices in combination with frequent family meals were associated with adolescents having a higher intake of fruit (*p* = 0.011). Latino fathers can have an important positive influence on adolescent dietary intake.

## 1. Introduction

Latino family strengths that can positively influence health outcomes include protective cultural values and beliefs, such as familism [1,2]. Familism is the core belief in the centrality of the family, which underlies the concepts of family connectedness, family involvement, and support [3]. Other protective factors are community and social support, including resources that address food security and healthy eating. Protective family strengths can contribute to strong relationships between parents and children, thus supporting positive parental involvement in adolescent food choices.

Studies have generally shown that poor eating habits were associated with excess weight gain and obesity among children and adolescents [4,5]. According to recent studies, Mexican-American and other Hispanic children had lower fruit and vegetable intakes [6,7,8], and higher sugar-sweetened beverage (SSB) [9], sweets/salty snacks [10], and fast food [11] intakes than recommended by the Dietary Guidelines for Americans (DGAs) [12] and other expert groups [13,14]. Current research shows that Hispanic children and adolescents had the highest obesity rates compared to other ethnic/racial groups, with about half of all Hispanic children and adolescents classified in the overweight or obese categories based on nationally representative data [15,16,17]. Residing in urban enclaves [18] and low-income households [19,20] was associated with obesity among multiethnic children and adolescents. Protective factors based on Latino family strengths can interact with risk factors to address health disparities among children and adolescents in urban, low-income households [3].

Both maternal and paternal caregivers have an important role in preventing childhood overweight and obesity through the formation of healthy food- and activity-related behaviors among youth [21,22]. Several reviews have identified a variety of food and activity parenting practices that influence adolescents’ food and activity behaviors, including setting expectations, role modeling, and managing availability [23,24]. Yet, studies have primarily focused on mothers when examining parental influence on adolescents’ diets and diet-related outcomes [21], with few studies addressing the influence of fathers [25,26,27,28,29]. The existing evidence base suggests that child and adolescent behaviors are influenced by their fathers’ food- and activity-related parenting practices [30,31,32]. Positive associations have been observed between fathers’ and children’s body weight and food intake [26,33]. In a cross-sectional sample of Canadian parents of children (5–12 years), relationships among food parenting practices were similar between mothers and fathers for some children’s eating behaviors but differentially associated with behaviors regarding food and satiety responsiveness [34]. For example, paternal restriction for weight practices, practices to accommodate the child, and use of covert control were associated with higher child food responsiveness, while only maternal restriction for weight practices were associated with higher food responsiveness.

Research among Latino families suggests that fathers play an important role in improving their children’s eating behaviors through food parenting practices and buying and preparing foods, and participating in meals [28,35,36]. A focus group study identified eight primary food and activity parenting practices reported by Latino fathers (n = 26) related to improving their children’s healthy lifestyles [35]. In another focus group study, Hispanic mothers reported that fathers had behaviors that supported healthy youth behaviors, including preparing healthy meals, using healthier cooking methods, shopping for healthy food with their children, and asking the child to participate in household chores and/or play sports [37]. Family meals have consistently been shown to positively influence the dietary behaviors of school-aged children and adolescents [38]. Studies suggested that having meals as a family played an important role in the formation of healthy eating habits among youth [38,39,40]. However, limited and inconsistent data were available on the relationship between family meal frequency and Latino children’s food behaviors and diet quality [41,42].

Environmental factors including fathers’ food parenting practices, fathers’ food/meal involvement, and family meals operate within the reciprocal determinism construct of Social Cognitive Theory to influence adolescents’ dietary behaviors [43,44]. In addition, Vaughn et al. [23] identified structure as one of the three fundamental constructs of food parenting practices. Within the structure construct, subconstructs included food parenting practices such as rules and limits (parents’ expectations), modeling, food availability, and meal and snack routines such as food preparation and having meals with family. More frequent family meals or fathers’ food/meal involvement may provide greater opportunities for fathers to communicate expectations, model healthy food intake, and influence the availability of healthy foods [38]. Studies have separately examined the impact of family meals and food parenting practices on the food behaviors of children in the general population [21,23,24,38,39,40]. However, interactions between paternal food parenting practices and supportive environmental factors such as family meals have only been examined to a limited extent [45].

The existing literature indicates an examination of associations among Latino fathers’ food parenting practices (i.e., setting expectations/limits, role modeling, making foods available), fathers’ food/meal involvement (planning meals, buying and preparing foods with the adolescent), frequency of family meals and adolescent dietary behaviors is warranted. Therefore, the purpose of this study was to test the hypothesis that separately or in combination, favorable Latino fathers’ food parenting practices, fathers’ food/meal involvement, and frequent family meals are associated with greater adolescent consumption of fruit and vegetables and lower consumption of sweets/salty snacks, sugary drinks, and fast food.

## 2. Materials and Methods

### 2.1. Study Design

This cross-sectional study used baseline survey data from a convenience sample of participants in a community-based intervention project (Padres Preparados, Jóvenes Saludables-Prepared Parents, Healthy Youth) at six community sites in the Minneapolis/St. Paul metropolitan area [46]. The randomized controlled intervention trial (identifier: NCT03641521) aimed to prevent overweight and obesity among Latino adolescents (10–14 years) by improving fathers’ food- and physical activity-parenting practices and youth energy balance-related behaviors. Social media, flyers, and announcements at community service centers and churches were used to recruit participants between September 2017 and February 2020. Fathers and adolescents provided consent and assent to participate in the study, respectively. Consent and assent forms explained all procedures involving educational and data collection sessions that fathers and adolescents were asked to complete in relation to the study objectives. Fathers and adolescents received separate cash compensation (USD 35 for fathers and USD 25 for adolescents) for their participation.

Baseline data were collected prior to randomization of participants into intervention and control groups. The intervention group attended 8-weekly educational sessions about nutrition, physical activity, and positive food and physical activity parenting practices. The 2.5 h educational sessions were conducted in-person at churches or community centers, with trained bilingual facilitators leading the interactive sessions. Evaluation data were collected in the same settings by trained research assistants at baseline, post, and 3 months after the intervention group educational sessions. The control group attended the same educational sessions after the 3-month data collection session. The data collection sessions lasted about 1 h per father/adolescent dyad including questionnaires, height and weight measurements, and dietary recall interviews. The study protocol was approved by the University of Minnesota Institutional Review Board (project identification code: 1511S80707).

### 2.2. Study Participants

Participants were Latino fathers or male caregivers with an adolescent (10–14 years) (n = 191 dyads). Eligibility criteria for fathers/caregivers were identifying as Latino, speaking Spanish, and having meals at least three times a week with their adolescent. Eligibility criteria for adolescents included being the child of a Latino father/caregiver and being 10–14 years of age. The intervention was intended to prevent overweight among those who were categorized as normal weight at baseline, prevent obesity among those categorized as overweight, and prevent severe obesity among those categorized as obese. At baseline, eligible fathers completed a self-administered survey in Spanish while eligible adolescents completed a self-administered survey in English and 24 h dietary recall interviews.

### 2.3. Sociodemographic and Household Characteristics of Participants

Fathers reported sociodemographic characteristics (age, education, employment, marital status, language spoken at home, and number of years in the US) and household characteristics (income, food security, and number of children in the home). Adolescents reported their own birthdate and sex.

Food security was measured by a combination of two questions from the USDA Food Security Survey Module [47]: “Within the past 12 months, we worried about whether our food would run out before we got money to buy more” and “Within the past 12 months, the food we bought just didn’t last and we didn’t have money to get more”. Response options for both questions were often true, sometimes true, and never true. If fathers responded “often” or “sometimes true” to one of the two questions, they were classified as food insecure. Prior research determined that these two questions had 78% sensitivity, 96% specificity, and convergent validity regarding the identification of food insecurity [47].

Language spoken at home was categorized as exclusively or primarily Spanish = 0, equally Spanish and English = 1, and more English than Spanish or only English = 2. Years in the US were classified according to four categories: <10 years = 0, ≥10–<20 years = 1, ≥20–<30 years = 2, and ≥30 years = 3.

### 2.4. Outcome Measures

#### 2.4.1. Adolescent Dietary Intake

To estimate dietary intake, 24 h dietary recall interviews were conducted using Nutrition Data System for Research software (NDSR; Nutrition Coordinating Center, University of Minnesota). The first recall was conducted in person during the baseline data collection session, with two additional recalls completed by phone within the next 1–2 weeks. The majority of adolescents (77%) completed at least two dietary recalls, with 53% completing three recalls. Recall interviews were balanced to reflect the distribution of weekdays and weekend days. Adolescents were asked to report all foods, beverages, and water they consumed in the last 24 h. A Food Amounts Booklet, which showed illustrations of foods or abstract shapes and figures in different sizes, was provided to assist in estimating amounts consumed. Intakes were averaged across the number of recalls per adolescent and reported as servings per day. Fruit and vegetable intakes were calculated separately using the NDSR fruit category total (excluding juice) and vegetable category total (excluding fried vegetables, fried potatoes, and white potatoes). SSB intake was calculated based on reported intake of beverages categorized by the NDSR software as sugar-sweetened beverages, which included sweetened soft drinks, fruit drinks, tea, coffee, coffee substitute, and water. Sweets/salty snack intake was calculated using foods from several NDSR categories, including chips and other salty snacks, meat and vegetable-based snacks, ready-to-eat cereals, grain-based desserts, dairy desserts, candies, sugars, jams, syrups, and sweet sauces. Intake of fast food type foods was calculated using foods from several NDSR categories, including fried vegetables, fried potatoes, and fried chicken, fish, and shellfish (commercial entrée and fast food).

#### 2.4.2. Adolescent Anthropometric Measurements

Adolescents’ body weight and height were measured separately twice in a private space using a digital scale (BWB-800 Scale, Tanita, IL, USA) and a stadiometer by a trained research assistant according to standardized procedures of the National Health and Nutrition Examination Survey (NHANES) [48]. BMI percentiles were generated by a SAS program for the 2000 CDC Growth Charts and categorized as underweight (<5th percentile), normal weight (5th–<85th percentile), overweight (85th–<95th percentile), and obese (≥95th percentile) [49].

#### 2.4.3. Fathers’ food Parenting Practices

Adolescents reported the perceived frequency of fathers’ food parenting practices (setting expectations/limits, role modeling, and making foods available at home) for intake of fruit, vegetables, SSBs, sweets/salty snacks, and fast food. High and low levels or frequencies for each parenting practice were created based on median values.

Adolescent-reported food parenting practice items and scales developed for this study were adapted from existing scales [50,51,52] and showed internal consistency for all scales based on Cronbach α coefficients >0.7 in a preliminary study [53]. Adequate criterion validity was demonstrated for 19 of the 21 parenting practice measures based on the adolescent report. Criterion validity was indicated by significantly higher adolescent-reported consumption of fruit and vegetables; lower consumption of SSBs, sweets/salty snacks, and fast foods; greater weekly physical activity hours; and fewer daily screen time hours among adolescents who reported high vs. low levels/frequencies of supportive parenting practices. However, father-reported parenting practice items and scales only showed criterion validity for 3 of the 21 parenting practice measures. These results indicated greater consistency between perceived frequency of paternal parenting practices and adolescent behaviors when adolescents reported parenting practice frequency vs. fathers. The percentage agreement between adolescent- and father-reported dichotomized responses varied from 49% to 68% for paternal expectations/limits, 51% to 70% for modeling, and 52% to 70% for availability practices. In general, adolescents reported lower frequencies of supportive food parenting practices than fathers. Adolescent-reported food parenting practices data were used in this study instead of father-reported data based on these preliminary testing results [53].

Setting expectations/limits. Adolescents’ perceptions of father expectations for fruit and vegetable intake were measured separately for fruits and vegetables by asking, “How many times in a day do you think your father wants you to eat [fruits, vegetables]?” Response options were 0 times or I don’t know, 1 time, 2 times, and 3 times or more. Adolescents’ perceptions of father limits for intake of sweets/salty snacks were assessed by asking, “How often does your father allow you to [drink SSBs, eat sweets/salty snacks, eat fast food]?” Response options were no [SSBs, sweets/salty snacks, fast food] are allowed, < 1 time/week, 1–3 times/week, 4–6 times/week, and one or more times/day, as often as I want, and I don’t know.

Role modeling. To evaluate adolescents’ perception of father role modeling of fruit and vegetable intake, adolescents were asked to report separately how many times in a week (1) “you see your father [eat fruit, vegetables, drink SSBs, eat sweets/salty snacks, eat fast food]?” and (2) “your father eats [fruit, vegetables, SSBs, sweets/salty snacks, fast food] with you?” Response options for each question were almost never or never, <1 time/week, 1–3 times/week, 4–6 times/week, and once a day or more. Responses to the two questions for each dietary behavior were coded from 1–5, summed and averaged to create a modeling score.

Making foods available at home. Adolescents’ perceptions of the frequency of father practices regarding home food availability/accessibility were assessed with three questions for each dietary behavior. Frequency of making fruit and vegetables available at home was measured by asking adolescents: “How often your father (1) buys (fruits, vegetables) (2) prepares (fruits, vegetables) for you to eat, and (3) makes a variety of (fruits, vegetables) available for you.” The frequency of making SSBs, sweets/salty snacks, and fast food available at home was determined by asking adolescents three questions: “How often does your father (1) buy (SSBs, sweets/salty snacks, fast food) for you to eat, (2) prepare (SSBs, sweets/salty snacks, fast food) for you to eat, and (3) give you money to buy (SSBs, sweets/salty snacks, fast food)?” Response options for each question were almost never or never, not often, sometimes, often, and almost always or always. The responses to the three questions for each dietary behavior were coded from 1–5, summed and averaged to create an availability score.

#### 2.4.4. Family Meals and Fathers’ Food/Meal Involvement

Frequency of family meals was assessed by asking adolescents: “During the past 7 days, how many times did you eat a meal with all or most of your family?” Response options included never, 1 to 2 times, 3 to 4 times, 5 to 6 times, 7 times, and more than 7 times [54]. Two frequency levels of family meals were created based on median values.

Fathers’ food/meal involvement was examined by asking adolescents three questions: “How often does your father plan meals together with you?” “How often does your father buy foods together with you?” and “How often does your father prepare foods together with you?” Five response options for each question (coded from 1–5) were almost never or never, not often, sometimes, often, and almost always or always. Responses of the three questions were averaged, and two levels of fathers’ food/meal involvement were created based on median values.

### 2.5. Statistical Analysis

All analyses were conducted using SAS 9.4 (Cary, NC, USA, 2002–2012). Descriptive statistics were performed to report results regarding adolescent and father sociodemographic characteristics, fathers’ food/meal involvement, family meals, and adolescent dietary intake. Cut-off points for fathers’ parenting practices, fathers’ food/meal involvement, and family meals were identified based on median analysis. The normality of adolescent dietary intake variables was assessed by visual examination of the histogram and Kolmogorov–Smirnov test.

Outliers for adolescent dietary intake data were examined using histograms and the interquartile range (IQR) formula. Intake data for a particular food group were removed from three adolescents based on values above (Q3 + 1.5 × IQR) and from one adolescent based on data entry error (a researcher entered an overestimation of fruit intake).

Adolescent dietary intake variables were not normally distributed; therefore, a square root transformation was used to approximate normality. For ease of interpretation of the results, non-transformed least-square means were presented with the p-values based on models including transformed variables.

Multiple linear regression models were used to examine adolescent intake (dependent variables) and fathers’ food/meal involvement and family meals variables alone and fathers’ food parenting practice variables alone (independent variables). Models were adjusted for adolescent age and sex and other sociodemographic father and/or adolescent characteristics based on results of preliminary comparison testing. Unstandardized coefficients represented the change in mean daily servings of adolescent dietary intake as a function of the independent variables.

Another set of regression models of adolescent dietary intake included fathers’ food/meal involvement, family meals, fathers’ food parenting practices, and their interactions. Models were examined using GLM (generalized linear mixed model) procedures adjusted for adolescent age and sex and sociodemographic variables based on preliminary comparison testing. For models with interactions (identified with a *p*-value < 0.10), the simple effects of each fathers’ food parenting practice within each fathers’ food/meal involvement and family meals category were calculated using slice statements and PLM (post GLM processing) procedures with Bonferroni corrections for multiple comparisons. Results of the models were presented as adjusted means and 95% CI. A *p*-value <0.05 was considered statistically significant.

## 3. Results

After screening for eligibility (n = 277) and accounting for those not attending baseline data collection sessions (n = 86), data from a total of 191 father/adolescent dyads were available for analysis for this study. Enrollment was evenly divided for boys and girls and approximately evenly divided for ages 10–11 years and 12–14 years (Table 1). One child was 8 years old during screening but turned 9 during the intervention, 2 others were 14 years old during screening but turned 15 during the intervention. Slightly more than half of the fathers were 41 years or older (56%), with 92% living with a spouse or partner. The distribution of fathers’ educational attainment showed that 20% had completed some college or more, 43% had a high school diploma or GED (General Education Development test that shows high school academic knowledge), and 37% had not completed high school. Approximately 75% of the fathers were employed full-time. The majority of the fathers reported yearly household income of ≤USD 49,999 (84%). The majority reported being food secure (63%) while 37% reported being food insecure. Most reported speaking exclusively or primarily Spanish at home (84%).

Nearly half of the adolescents (47%) reported having family meals ≥7 times a week vs. ≤6 times a week (Table 1). About half (49%) reported that their father was involved often or always vs. never to sometimes in planning meals, buying, and preparing foods with them. The majority of adolescents (58%) were classified in the overweight or obese category.

### 3.1. Sociodemographic Differences in Adolescent Dietary Intake

SSB intake among adolescents with fathers who participated in financial assistance programs (mean = 0.3 serving/day, SD = 0.4) was lower compared to those with fathers who did not participate (mean = 0.5, SD = 0.7, *p* = 0.008). Fast food intake by adolescents with a single father (mean = 0.9 serving/day, SD = 1.0) was higher than those with a married father (mean = 0.4, SD = 0.8, *p* = 0.020). No other differences were observed by the remaining fathers’ or households’ demographic characteristics (data not shown).

### 3.2. Associations among Family Meals, Fathers’ Food/Meal Involvement, Fathers’ Food Parenting Practices, and Adolescent Dietary Intake

Based on adjusted linear regression models, no significant associations were observed among the frequency of family meals, fathers’ food/meal involvement, and adolescent intakes of fruit, vegetables, SSBs, sweets/salty snacks, and fast food (data not shown).

Adjusted models indicated that adolescent fruit intake was higher when fathers made fruits available at home more often (β = 0.45, *p* = 0.011) (Table 2). Adolescent vegetable intake was higher when fathers modeled intake of vegetables more often and made vegetables available at home more often (β = 0.59, *p* = 0.002 and β = 0.41, *p* = 0.021, respectively).

Adolescent SSB intake was lower when fathers set lower limits for SSB intake and fathers made SSBs available at home less often (β = −0.19, p = 0.025 and β = −0.17, *p* = 0.037, respectively) (Table 3). Adolescent intake of sweets/salty snacks was lower when fathers modeled intake of sweets/salty snacks less often (β = −0.93, *p* = 0.001) and when fathers made sweets/salty snacks available at home less often (β = −0.61, *p* = 0.013). Adolescent fast food intake was lower when fathers set lower limits for fast food intake (β = −0.37, *p* = 0.015).

### 3.3. Interactive Associations of Fathers’ Food/Meal Involvement, Family Meals, and Fathers’ Food Parenting Practices on Adolescent Dietary Intake

A notable interaction was observed based on the regression model with adolescent fruit intake, family meal frequency and father fruit expectations (*p* < 0.10). Adolescent fruit intake was significantly higher when fathers set higher expectations for fruit intake and family meals were more frequent vs. less frequent (1.2 fruit servings/day, CI = [0.80, 1.57] vs. 0.5 fruit servings/day, CI = [0.24, 0.85], *p* = 0.011).

An interaction was also observed for sweets/salty snacks intake, fathers’ food/meal involvement, and fathers making sweets/salty snacks available at home (*p* < 0.10). Adolescent intake of sweets/salty snacks was significantly lower when fathers were involved less often in planning meals, buying and preparing foods with the adolescent (unfavorable) and when fathers made sweets/salty snacks available at home less vs. more often (favorable) (0.9 sweets/salty snack servings/day, CI = [0.57, 1.22] vs. 1.9 sweets/salty snack servings/day, Cl = [1.43, 2.44], *p* = 0.001).

An interaction was observed with adolescent SSB intake, fathers’ food/meal involvement, and father SSB modeling (*p* < 0.10). Adolescent SSB intake was significantly lower when fathers modeled intake of SSBs more often (unfavorable) and when fathers were involved less vs. more often in planning meals and buying and preparing foods with the adolescent (unfavorable) (0.1 SSB servings/day, CI = [0.02, 0.25] vs. 0.4 SSB servings/day, CI = [0.21, 0.59]. *p* = 0.017). Interactions from the remaining models examined did not yield *p* < 0.10.

## 4. Discussion

This cross-sectional study examined the influence of Latino fathers’ food parenting practices, family meals, and fathers’ food/meal involvement on adolescent fruit, vegetable, SSB, sweets/salty snack, and fast food consumption separately and in combination. When examined separately, Latino fathers’ food parenting practices were associated with higher healthy food intake and lower unhealthy food intake by adolescents, similar to the general parenting practice literature [32,45,55]. However, Latino fathers’ food/meal involvement and family meal frequency were not independently associated with adolescent food intake, which was unexpected. When interactions were examined, a limited number of associations were observed among adolescent dietary intake and combinations of family meal frequency, fathers’ food/meal involvement and fathers’ food parenting practices.

Parents may encourage healthy dietary behaviors among adolescents by providing healthy foods at home and being positive role models by eating foods such as fruits and vegetables with their adolescent. Previous studies involving predominantly mothers showed that parental modeling and home food availability were associated with youth dietary intake [55,56,57,58,59]. The current study lends continued support to those findings by also showing that father modeling and making foods available predicted adolescents’ dietary consumption. Latino fathers’ modeling and making foods available may have been associated with adolescent consumption in part because Latino fathers have increased their level of involvement in food preparation at home over time [60]. Taillie et al. (2018) [60] showed in a U.S. nationally representative sample that from 2003 to 2016, the proportion of Hispanic fathers who cooked at home increased from 31.2% to 41.6%, and so did time spent cooking. In the current study, about half of the adolescents reported that their fathers were often or always involved in planning, buying, and preparing foods with them.

The current findings showed that adolescent fast food intake was lower when adolescents perceived that their fathers set limits for fast food intake for them. In contrast, Latino mothers reported that fathers had unsupportive food behaviors for children, including bringing high-calorie foods (e.g., pizza) and sugary drinks into the home [37]. NHANES 2011–2016 data showed that Hispanic/Latino men worked more outside the home and were more likely to eat meals away from home compared to women [61]. Findings from the current study indicated that as reported by adolescents, Latino fathers may have recognized the need to limit fast food intake for their adolescents, while other studies indicate that time constraints and environmental conditions may not always support favorable paternal parenting practices. However, the youth in the current study were interested in enrolling in a father/adolescent nutrition and physical activity health intervention, and therefore may not have had similar perceptions regarding their fathers’ limits on fast food intake as a general Latino adolescent population.

The frequency of meals as a family has generally been related to adolescents’ food choices [38,39,62] with some evidence indicating that it may also be related to dietary behaviors of Latino adolescents [41,63]. The findings from the current study did not show an independent association between the frequency of family meals and adolescents’ dietary intake. Similarly, a previous study involving urban Hispanic adolescents (10–14 years) did not show an association between family meal frequency and youth Healthy Eating Index (HEI) scores [42]. However, the frequency of family meals while watching TV was associated with lower HEI scores [42], indicating that other habits in conjunction with family meals may exert an influence on youth food choices and diet quality. Other contextual factors, such as TV viewing during family meals, were not examined in the current study.

The present study found that Latino adolescents had higher fruit intake when the combination of adolescent perceptions of fathers setting higher expectations for fruit intake and frequent family meals were considered. The cultural value of family connectedness among Latino families, which promotes the frequency of family meals, may have contributed to opportunities where fathers could engage in favorable parenting practices. The findings from the current study are further supported by a study with urban adolescents (including ~34% Hispanic adolescents), which showed a strong association between greater FV intake and frequent family meals (≥5 times/week) in combination with favorable food parenting practices [45].

The results from the current study indicated that Latino adolescent consumption of sweets/salty snacks was lower when adolescents perceived that their fathers modeled intake of sweets/salty snacks less often and made sweets/salty snacks available at home less often. These findings are consistent with a previous study showing that adolescent intake of sweets/salty snacks was lower when parents had favorable modeling practices [55] and another study indicating that less frequent home snack availability was associated with lower snack intake among children [64]. Snacking constitutes about one-third of the daily energy intake among US children and adolescents, with desserts, salty snacks, and SSBs as the major sources of calories from snacks [65]. According to the findings from dietary recalls of eight nationally representative surveys from 1977 to 2014, average energy intake from snacks among Mexican-American children aged 2–18 significantly increased from 205 to 453 kcals per snacking occasion [10]. Nationally representative data from 1977 to 2012 also showed that average energy intake from snacks per Hispanic adult significantly increased from 167 to 418 kcals per day [66]. Based on adolescent perceptions from the current study, Latino fathers’ engagement in parenting practices that lower adolescent sweets/salty snack intake may be an important intervention target to decrease intake of energy-dense, nutrient-poor foods consumed as snacks.

The current study also showed that adolescent intake of sweets/salty snacks was lower when adolescents perceived that fathers made snacks available at home less often but were less often involved in planning meals, buying, and preparing foods. The extent of fathers’ involvement in the meal planning/food preparation process was shown to have increased over time [60], but it still might be related to traditional Latino gender roles, which view mothers as being responsible for cooking meals while fathers support their families financially [67]. A qualitative study showed that Latino fathers reported doing less food work, including buying foods, cooking, and dietary decision-making for the family compared to mothers [68]. Another study used data from two linked population-based studies with diverse parents and adolescents and showed that the proportion of fathers who reported they “usually” prepared food for the family was lower than mothers for all races/ethnicities [69]. However, the findings from the current study showed that fathers can still improve their children’s dietary intake by not making these foods available at home regardless of which parent is most often involved in the meal planning/food preparation process.

The strengths of this study include its large population-based sample of Latino adolescents and their fathers or male caregivers, the comprehensive examination of adolescent dietary intake, and ability to examine associations with multiple fathers’ food parenting practices in conjunction with family meal frequency and fathers’ food/meal involvement. Another strength was the use of adolescent-reported paternal parenting practice frequency, which was shown to have better criterion validity with adolescent behaviors than father report in a preliminary study [53]. This study had several limitations, which should be considered when interpreting the results. Errors in adolescent dietary intake data may have resulted from poor recall or social desirability leading to over-reporting fruit and vegetable intakes and under-reporting intake of sweets/salty snacks, SSBs, and fast food. Additionally, the findings may not be generalizable to the broader Latino population in the US because study participants were recruited from a limited geographical area, and most were from low-income families. Furthermore, two study sites were community centers that regularly offered other health and nutrition classes; thus, the fathers and youth who agreed to participate in the study might have enrolled due to their interest in nutrition and health, which makes them different from the general population. Monetary incentives may also have influenced decisions to enroll in the study. Finally, data on adolescent perceptions of mothers’ parenting practices and food/meal involvement were not collected; therefore, these findings could not be shared or integrated into the discussion.

## 5. Conclusions

The current study addressed a gap in the literature by examining how Latino fathers’ parenting practices, food/meal involvement, and family meals were separately and in combination related to dietary intake among Latino adolescents. Understanding how fathers’ parenting practices influence adolescent intake may be improved if other contextual factors such as eating as a family are also considered. These findings emphasize the importance of including fathers in behavioral and family-based research to prevent overweight and obesity among adolescents. Interventions intended to improve adolescent diet and health may become more effective by including a focus on fathers’ parenting practices and family meals.

## Figures and Tables

**Table 1 ijerph-18-08226-t001:** Adolescent/father dyad sociodemographic characteristics and adolescent perceptions of family meals, fathers’ food/meal involvement, and adolescent dietary intake (*n* = 191).

Participants’ Characteristics		*n* (%) ^a^ or Mean ± SD
Adolescent demographic characteristics		
Sex		
	Male	92 (49)
	Female	96 (51)
Age		
	10–11 years	94 (50)
	12–14 years	95 (50)
Father demographic characteristics		
Age		
	20–≤40 years	84 (44)
	≥41 years	105 (56)
Education		
	Middle school or lower	70 (37)
	GED or high school	80 (43)
	Some college or higher	37 (20)
Employment		
	Self-employed	27 (15)
	Unemployed	7 (4)
	Part-time employed	12 (6)
	Full-time employed	136 (75)
Marital status		
	Single	14 (8)
	Married or with a partner	172 (92)
Household income		
	* ≤USD 24,999*	73 (40)
	*USD 25,000–≤USD 49,999*	80 (44)
	* USD 50,00–USD 99,999*	29 (16)
Language spoken at home		
	Exclusive or primarily Spanish	157 (84)
	Equally Spanish and English	27 (14)
	More English than Spanish or only English	3 (2)
Food security ^b^		
	Food secure	120 (63)
	Food insecure	71 (37)
Number of children in the home		2.6 ± 1.2
Number of years in the US		19.2 ± 6.5
Participation in financial assistance programs ^c^		
	≥1 time	53 (29)
	Never	130 (71)
Family meals ^d^		
	Less often	100 (53)
	More often	90 (47)
Fathers’ food/meal involvement ^e^		
	Less often	96 (51)
	More often	94 (49)
Adolescent intake ^f^		
	Fruit, servings/day	1.1 ± 1.3
	Vegetable, servings/day	1.5 ± 1.2
	SSB, servings/day	0.5 ± 0.7
	Sweets/salty snacks, servings/day	1.7 ± 1.4
	Fast food, servings/day	0.4 ± 0.8
Adolescent BMI group		
	Underweight: <5th percentile	2 (1)
	Normal weight: 5th–<85th percentile	74 (41)
	Overweight: 85th–<95th percentile	46 (26)
	Obese: ≥95th percentile	58 (32)

^a^ All data were reported by adolescents except for fathers’ demographic characteristics. ^b^ Food security was determined by combining responses to two questions from the USDA Food Security Module about the food eaten in the participant’s household and whether they were able to afford the food they needed. ^c^ WIC, SNAP-Ed, free or reduced meals at school, and the Minnesota Family Investment Program. ^d^ Frequency responses for family meals questions were never, 1 to 2 times, 3 to 4 times, 5 to 6 times, 7 times, and more than 7 times in a week and categorized as less often (≤5–6 times/week) and more often (≥7 times/week). ^e^ Frequency responses for three fathers’ food/meal involvement (father involvement in planning, buying, and preparing foods with their adolescent) questions were almost never or never = 1, not often = 2, sometimes = 3, often = 4, and almost always or always = 5). Responses were averaged and categorized as less often (≤3) and more often (>3). ^f^ Intake based on adolescent 24 h dietary recalls using NDSR software.

**Table 2 ijerph-18-08226-t002:** Adjusted associations between adolescent-reported fruit and vegetable intake and fathers’ food parenting practices (*n* = 191) ^a^.

Fruit Parenting Practices	Unstandardized Regression Coefficients (95% CI) for Fruit Intake ^b^	Vegetable Parenting Practices	Unstandardized Regression Coefficients (95% CI) for Vegetable Intake ^b^
Setting Expectations ^c^		Setting Expectations ^c^	
Low intake	Ref.	Low intake	Ref.
High intake	0.25 (−0.18, 0.68)	High intake	0.13 (−0.32, 0.58)
Role modeling ^d^		Role modeling ^d^	
Less often	Ref.	Less often	Ref.
More often	0.18 (−0.21, 0.56)	More often	0.59 (0.23, 0.95) **
Making available at home ^e^		Making available at home ^e^	
Less often	Ref.	Less often	Ref.
More often	0.45 (0.07, 0.82) *	More often	0.41 (0.05, 0.78) *

^a^ Between-group comparisons were conducted using multiple linear regression analyses, * *p*-value < 0.05, ** *p*-value < 0.01. ^b^ All models were adjusted for adolescent age and sex. ^c^ Setting expectations for fruit and vegetable intake was based on one item each assessing how frequently fathers wanted their adolescents to eat (fruit, vegetables) in a day: 0 times, 1 time, 2 times, 3 times or more, and I don’t know (I don’t know option was treated as missing). Expected intake levels were low intake ≤2 times/day and high intake ≥3 times/day. A high intake level was considered favorable. ^d^ Role modeling was based on the average of two items, each assessing how many times adolescents saw their father eat these foods and how many times their father ate these foods with them: almost never or never, <1 time/week, 1–3 times/week, 4–6 times/week, and once a day or more. Fruit and vegetable modeling levels were less often (≤1–3 times/week) and more often (≥4–6 times/week). Modeling intake of fruits and vegetables more often were considered favorable. ^e^ Making fruit and vegetables available at home was based on the average of three items, each assessing frequency of fathers buying, preparing, and making sure adolescents have different kinds: almost never or never = 1, not often = 2, sometimes = 3, often = 4, and almost always or always = 5. Fruit availability levels were less often (<3.6) and more often (≥3.6). Vegetable availability levels were less often (<3.3) and more often (≥3.3). Making fruits and vegetables available at home more often was considered favorable.

**Table 3 ijerph-18-08226-t003:** Adjusted associations between adolescent-reported sweets/salty snack, SSB, and fast food intakes and fathers’ food parenting practices (*n* = 191) ^a^.

SSB ^b^ Parenting Practices	Unstandardized Regression Coefficients (95% CI) for SSB Intake ^c^	Sweets/Salty Snacks Parenting Practices	Unstandardized Regression Coefficients (95% CI) for Sweets/Salty Snack Intake ^c^	Fast-Food Parenting Practices	Unstandardized Regression Coefficients (95% CI) for Fast Food Intake ^c^
Setting Limits ^d^		Setting Limits ^d^		Setting Limits ^d^	
Low intake	−0.19 (−0.40, 0.03) *	Low intake	−0.33 (−0.91, 0.26)	Low intake	−0.37 (−0.65, −0.10) *
High intake	Ref.	High intake	Ref.	High intake	Ref.
Role modeling ^e^		Role modeling ^e^		Role modeling ^e^	
Less often	−0.00 (−0.21, 0.20)	Less often	−0.93 (−1.45, −0.42) **	Less often	−0.10 (−0.38, 0.19)
More often	Ref.	More often	Ref.	More often	Ref.
Making available at home ^f^		Making available at home ^f^		Making available at home ^f^	
Less often	−0.17 (−0.38, 0.04) *	Less often	−0.61 (−1.05, −0.18) *	Less often	−0.13 (−0.39, 0.12)
More often	Ref.	More often	Ref.	More often	Ref.

^a^ Between-group comparisons were conducted using multiple linear regression analyses, * *p*-value < 0.05, ** *p*-value < 0.01. ^b^ SSB = Sugar-sweetened beverages. ^c^ All models were adjusted for adolescent age and sex. Models with adolescent SSB intake and SSB parenting practices were also adjusted for household participation in financial assistance programs; models with adolescent sweets/salty snack intake and sweets/snack parenting practices, and fast food intake and fast food parenting practices were also adjusted for fathers’ marital status. ^d^ Setting limits for SSBs, sweets/salty snacks, and fast-food intakes were based on one item assessing how frequently fathers allowed their adolescent to [drink SSBs, eat sweets/salty snacks, eat fast food] with response options: no [SSBs, sweets/salty snacks, fast food] are allowed, <1 time/week, 1–3 times/week, 4–6 times/week, and one or more times/day, as often as I want, and I don’t know (I don’t know was treated as missing). Expected intake levels for SSBs and sweets/salty snack intake were low intake (≤1–3 times/week) and high intake (≥4–6 times/week). Expected intake levels for fast food intake were low intake (less than 1 time/week) and high intake (≥1–3 times/week). A low expected intake level for SSBs, sweets/salty snacks and fast food was considered favorable. ^e^ Role modeling was based on the average of two items, each assessing how many times adolescents saw their father eat these foods and how many times their father ate these foods with them with response options: almost never or never, <1 time/week, 1–3 times/week, 4–6 times/week, and once a day or more for SSBs, sweets/salty snacks. Levels of role modeling for SSBs, sweets/salty snacks, and fast food were less often (<1 time/week) and more often (≥1–3 times/week). Modeling intake of SSBs, sweets/salty snacks, and fast food less often were considered favorable. ^f^ Making sweets/salty snacks, SSBs, and fast food available at home was based on the average of three items assessing frequency of fathers buying, preparing, and giving adolescents money to buy with response options: almost never or never = 1, not often = 2, sometimes = 3, often = 4, and almost always or always = 5. Levels of availability were less often (<2) and more often (≥2). Making SSBs, sweets/salty snacks, and fast food available less often was considered favorable.

## Data Availability

The de-identified data used in this study are available on request from the corresponding author. The data are not publicly available because the data analysis phase of the intervention portion of the study has not been completed.

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
