# Peer review of "Adolescent-Reported Latino Fathers’ Food Parenting Practices and Family Meal Frequency Are Associated with Better Adolescent Dietary Intake"

_ijerph, 2021, doi:10.3390/ijerph18158226_

Round 1

Reviewer 1 Report

Thank you for the opportunity to review the submitted manuscript. There is a tremendous lack of research examining the role of fathers on their children’s energy-balance behaviors. Thus, this study sought to examine the relationships among adolescent-reported paternal parenting practices (i.e., involvement in planning meals, buying/preparing foods, and family meal frequency), separately and in combination, with adolescent food intake. While there are numerous strengths with the current manuscript, interpretation of findings requires some adjustments as indicated. Some additional edits are also noted.

Overall:

When relaying information about identifying relationships among multiple variables (i.e., more than two), the word “between” should not be used; rather, “among” should be used. For example, on page 2, line 91, the following statement, “The existing literature indicates an examination of associations between Latino father food parenting practices (i.e., setting expectations/limits, role modeling, making foods available), father food/meal involvement (planning meals, buying and preparing foods with the adolescent), frequency of family meals and adolescent dietary behaviors is warranted,” should be changed to, “The existing literature indicates an examination of associations among Latino father food parenting practices (i.e., setting expectations/limits, role modeling, making foods available), father food/meal involvement (planning meals, buying and preparing foods with the adolescent), frequency of family meals and adolescent dietary behaviors is warranted.”

Abstract:

Failure to identify adolescents as the reporters of fathers’ behaviors is misleading. This information should be provided in the abstract, and perhaps in the title as well.

Discussion:

Page 10, Line 384: “Latino fathers recognized the need to limit fast food intake” cannot be stated for this study as fast food intake was solely reported by adolescents, and not their fathers. Please rephrase so that this finding is presented in light of adolescent reporting, rather than reporting directly from the fathers themselves.

Page 10, Line 398: Similarly, the statement, “Latino adolescents had higher fruit intake when the combination of fathers setting higher expectations for fruit intake…” also needs to account for adolescent reporting of fathers’ behaviors, rather than reporting directly from the fathers themselves.

Page 10, Line 407: Again, the following statement, “adolescent consumption of sweets/salty snacks was lower when fathers modeled intake…” needs to account for adolescent reporting of fathers’ behaviors, rather than reporting directly from the fathers themselves.

Page 10, Line 421: Lastly, the statement, “Adolescent intake of sweets/salty snacks was lower when fathers made snacks available at home less often but were less often involved in planning meals…” needs to account for adolescent reporting of fathers’ behaviors, rather than reporting directly from the fathers themselves.

While the authors addressed the interaction among adolescent fruit intake, family meal frequency, and father fruit expectations, along with the interaction among adolescent sweets/salty snacks intake, father food/meal involvement, and availability of snacks in the home, they failed to discuss the interaction found among adolescent SSB intake, father food/meal involvement, and father SSB modeling.

Another limitation that should be noted is the lack of father responses for the independent variables measured. It is possible that children’s interpretations of their fathers’ actions may not be similar to those reported by their fathers.

Author Response

Dear reviewer,

Thank you very much for the constructive feedback. We addressed your comments point by point below.

Reviewer 1

Thank you for the opportunity to review the submitted manuscript. There is a tremendous lack of research examining the role of fathers on their children’s energy-balance behaviors. Thus, this study sought to examine the relationships among adolescent-reported paternal parenting practices (i.e., involvement in planning meals, buying/preparing foods, and family meal frequency), separately and in combination, with adolescent food intake. While there are numerous strengths with the current manuscript, interpretation of findings requires some adjustments as indicated. Some additional edits are also noted.

Comment 1: Overall: When relaying information about identifying relationships among multiple variables (i.e., more than two), the word “between” should not be used; rather, “among” should be used. For example, on page 2, line 91, the following statement, “The existing literature indicates an examination of associations between Latino father food parenting practices (i.e., setting expectations/limits, role modeling, making foods available), father food/meal involvement (planning meals, buying and preparing foods with the adolescent), frequency of family meals and adolescent dietary behaviors is warranted,” should be changed to, “The existing literature indicates an examination of associations among Latino father food parenting practices (i.e., setting expectations/limits, role modeling, making foods available), father food/meal involvement (planning meals, buying and preparing foods with the adolescent), frequency of family meals and adolescent dietary behaviors is warranted.”

Response 1:  We replaced the word “between” with “among” on lines 104 and throughout the paper where we describe relationships among multiple variables.

Abstract:

Comment 2: Failure to identify adolescents as the reporters of fathers’ behaviors is misleading. This information should be provided in the abstract, and perhaps in the title as well.

Response 2: We revised the title as follows:

Adolescent-reported Latino father food parenting practices and family meal frequency are associated with better adolescent dietary intake.

We had indicated in the abstract that data were from adolescents (line 23). We had also indicated that data on frequency of father food parenting practices, fathers' food/meal involvement, and family meals were reported by adolescents (line 26). We added that fathers reported sociodemographic characteristics (line 24).

Discussion:

Comment 3: Page 10, Line 384: “Latino fathers recognized the need to limit fast food intake” cannot be stated for this study as fast food intake was solely reported by adolescents, and not their fathers. Please rephrase so that this finding is presented in light of adolescent reporting, rather than reporting directly from the fathers themselves.

Response 3: We explained why we used adolescent vs. father-reported data on father food parenting practices in the Father food parenting practices section of the methods (lines xxx) as follows:

However, father-reported parenting practice items showed limited criterion validity, and significant discrepancies were identified between adolescents’ and fathers’ reports of parenting practices. Therefore, adolescent reported food parenting practices data were used in this study.

We revised the statement to indicate that data on father food parenting practices were reported by adolescents as follows (lines 426-430):

Findings from the current study indicated that as reported by adolescents, Latino fathers may have recognized the need to limit fast food intake for their adolescents, while other studies indicate that time constraints and environmental conditions may not always support favorable paternal parenting practices.

We also revised in lines 421-422 as follow:

The current findings showed that adolescent fast food intake was lower when adolescents perceived that their fathers set limits for fast food intake for them.

Comment 4: Page 10, Line 398: Similarly, the statement, “Latino adolescents had higher fruit intake when the combination of fathers setting higher expectations for fruit intake…” also needs to account for adolescent reporting of fathers’ behaviors, rather than reporting directly from the fathers themselves.

Response 4: This statement was revised as follows (lines 445-447):

The present study found that Latino adolescents had higher fruit intake when the combination of adolescent perceptions of fathers setting higher expectations for fruit intake and frequent family meals were considered.

Comment 5: Page 10, Line 407: Again, the following statement, “adolescent consumption of sweets/salty snacks was lower when fathers modeled intake…” needs to account for adolescent reporting of fathers’ behaviors, rather than reporting directly from the fathers themselves.

Response 5: This statement was revised as follows (lines 454-456):

The results from the current study indicated that Latino adolescent consumption of sweets/salty snacks was lower when adolescents perceived that fathers modeled intake of sweets/salty snacks less often and made sweets/salty snacks available at home less often.

Comment 6: Page 10, Line 421: Lastly, the statement, “Adolescent intake of sweets/salty snacks was lower when fathers made snacks available at home less often but were less often involved in planning meals…” needs to account for adolescent reporting of fathers’ behaviors, rather than reporting directly from the fathers themselves.

Response 6: This statement was revised as follows (lines 472-474):

The current study also showed that adolescent intake of sweets/salty snacks was lower when adolescents perceived that fathers made snacks available at home less often but were less often involved in planning meals, buying, and preparing foods.

Comment 7: While the authors addressed the interaction among adolescent fruit intake, family meal frequency, and father fruit expectations, along with the interaction among adolescent sweets/salty snacks intake, father food/meal involvement, and availability of snacks in the home, they failed to discuss the interaction found among adolescent SSB intake, father food/meal involvement, and father SSB modeling.

Response 7: After an additional review of relevant literature, we were unable to explain the interaction among two unfavorable conditions (adolescents reporting that fathers modeled intake of SSBs more often and that fathers were less often involved in planning meals and buying and preparing foods with the adolescent) and a favorable outcome (lower adolescent SSB intake). We did not feel that discussing the lack of explanation for the interaction, or that more research is needed on the topic would be helpful, therefore, we choose not to add discussion about this finding.

Comment 8: Another limitation that should be noted is the lack of father responses for the independent variables measured. It is possible that children’s interpretations of their fathers’ actions may not be similar to those reported by their fathers.

Response 8: We had previously indicated in the methods section 2.3 (lines 148-150) that fathers reported sociodemographic characteristics (independent variables) and adolescents reported their own birthdate and sex (lines 150-151).

Fathers reported sociodemographic characteristics (age, education, employment, marital status, language spoken at home, and number of years in the US) and household characteristics (income, food security, and number of children in the home). Adolescents reported their own birthdate and sex.

We also added a sentence to the abstract to clarify (lines 24-25):

Fathers reported sociodemographic characteristics.

Reviewer 2 Report

The authors of this interesting study analyse the influence of Latino father food parenting practices on adolescents intake. The studt design, methods and data analysis are adequate. The text is well organized and clearly written.

Minor comments:

Methods section:

page 3, lines 147-148

The authors describe they  provided participants a food amounts booklet.

What kind of booklet: models? pictures? household measures?

page 4, line 153.

Did you consider milk based sugar sweetened beverages, such as milk-shakes or drinking yoghourt?

Author Response

Dear reviewer,

Thank you very much for the constructive feedback. We addressed your comments point by point below.

Reviewer 2

The authors of this interesting study analyze the influence of Latino father food parenting practices on adolescents intake. The study design, methods and data analysis are adequate. The text is well organized and clearly written.

Minor comments:

Comment 1: Methods section:

page 3, lines 147-148

The authors describe they provided participants a food amounts booklet.

What kind of booklet: models? pictures? household measures?

Response 1: We described the food amount booklet in the methods (lines 174-176) as follows:

A Food Amounts Booklet, which showed illustrations of actual food or abstract shapes and figures in different sizes, was provided to assist in estimating amounts consumed.

Comment 2: page 4, line 153.

Did you consider milk based sugar sweetened beverages, such as milk-shakes or drinking yoghourt?

Response 2: The Nutrition Data System for Research categorized milk-based beverages as dairy alternatives rather than sugar sweetened beverages. Thus, we did not consider milk-based sugar sweetened beverages as sugar sweetened beverages and did not revise the text.

Reviewer 3 Report

Thank you for the opportunity to review this paper.  It's interesting to consider paternal influences and it is very nice to look at parent / child dyad data.  The comments below pertain mainly to methodology and greater application of results in the discussion, and exploration of limitations.  

Abstract

GLM & PLM – use full explanations.

‘Associated with…’ in results– provide some figures for the abstract

Introduction

Line 45 and 49 – past tense is used to describe findings and it would be better to complement with year of data collection /publication for context or use present tense …as in ‘current research shows that…’

Line 52-54            reference this statement please

Line 57                  “several reviews and a qualitative study…” – it would be best to summarise review findings and separately discuss context using qualitative studies as do not think appropriate to aggregate here.

Line 62-63            Expand this to describe (if known) any unique role of the father (as distinct from mother) 

Line 78-79            How is it in line with social cognitive theory exactly?

Methods

A RCT is mentioned in the methods, however no details are given about this.  As per line 117, it seems this paper presents data from intervention and control groups at baseline before randomization?  That needs to be stated more clearly.  We also need a bit of detail about what the intervention involves to provide context as to why they might be participating.  Also personnel involved in the study, modality and location, frequency, duration etc.

What other study measures were taken given their inclusion in a wider intervention.  Did you have anthropometric data for the sample?  And if yes, can these be used to assess the accuracy of reported energy intake from the 24-hr recall?  I think it would be important to know if the proportion of the sample defined as healthy weight/overweight/obese.  The aim of the study is stated to be 'prevention of overweight and obesity' which implies that possibly they were not overweight/obese sample, but this needs clarification.  It would also enhance the methods section to know in what context were the dietary behaviour data collected - alongside other questionnaires and measures?

What did participants know about the study from informed consent?  Were all study objectives clear to participants, both fathers and children?

Line 141-142      should it be ‘analysed’ instead of ‘completed’?  I’m not sure I understand the structure of what is data collection and what is analysis in this paragraph

Line 152                SSB – is it ‘categorised’ rather than ‘calculated’?  Similarly line 154 and 157 .  I’m not sure I understand, again, the difference between data collection, analysis and calculations here, it needs to be much clearer.

Lines160-163      I don’t think IQR needs to be explained.  However, this paragraph needs to clearer – explain data entry error and also were all these adolescents’ data removed or just for certain food groups?  Would removing participants showing outlier data be more appropriate for statistical consistency?  Suggest integrating this with statistical analysis for consistency in reporting methods.

Line 165                For adolescents reporting the perceived frequency of father food parenting practices -   What tool was used – developed for this study?  Any references that informed its development?  Who administered the questions (qualification, training?), in what setting and how long did it take?

Line 206-211       If father did not plan, buy, or prepare food with the adolescent, how was fathers involvement in meals comprehensively accounted for?  For example, they planned, bought, or prepared food for the family without direct child involvement?  Is that what was being asked in these questions?

Results

Line 249                What is GED?

Line 247                would it be clearer to say ‘living with a spouse or partner?’

Line 251                37% or over one third being food insecure seems like a more notable finding and perhaps worth focusing on their characteristics here.

Line 253-255       Phrasing confusing here, ‘vs. less often’  - state more clearly.

Discussion

Lines 372-377     what was known about father involvement in food planning, shopping, and preparation among your sample, as reported by fathers?  Good to include this alongside discussion here and if not known, important to acknowledge as limitation.  Anything known about mothers or partners that could be shared in the findings for this study or integrated into the discussion.

Line 378-386       the sample within the study were different to general Latino community as they were already enrolled on a father/child health intervention – how does that influence this aspect of the discussion?

Line 418-420      Is there anything known about adult male Latino snacking (similar to child data) from national data?

Line 421-423       Rephrase this sentence to ensure relevance of latter element of the statement is clear

Limitations          It is a very decent sample size, however a ‘large population-based sample’ needs further explanation as in the methods it is a convenience sample, therefore likely biased in certain directions (as also alluded to).    It would be good to include more context about recruitment and what kind of selection biases were noted for the sample.  The (possible) influence of the monetary incentive also needs discussion in the limitations.  Many potential confounding factors are not acknowledged in the limitations for this study and need to be considered, what is not known about the sample? – e.g. information about maternal/partner practices and how these explain or influence family food behaviours (for both father and child).  Have any studies looked at maternal and paternal practices together and if yes, what can these types of studies tell us?    

Author Response

Dear reviewer,

Thank you very much for the constructive feedback. We addressed your comments point by point below.

Reviewer 3

Thank you for the opportunity to review this paper. It's interesting to consider paternal influences and it is very nice to look at parent / child dyad data. The comments below pertain mainly to methodology and greater application of results in the discussion, and exploration of limitations.

Abstract

Comment 1: GLM & PLM – use full explanations.

Response 1: We explained GLM and PLM in the abstract (in lines 28-29) and statistical analysis section as follows:

Abstract: The analysis included regression models using GLM (generalized linear mixed model) and PLM (post GLM processing) procedures (lines 28-29).

Statistical analysis: Models were examined using GLM (generalized linear mixed model) procedures adjusted for adolescent age and sex and sociodemographic variables based on preliminary comparison testing. For models with interactions (identified with a p-value < 0.10), the simple effects of each father food parenting practice within each father food/meal involvement and family meals category were calculated using slice statements and PLM (post GLM processing) procedures with Bonferroni corrections for multiple comparisons (lines 275-280).

Comment 2: ‘Associated with…’ in results– provide some figures for the abstract

Response 2: We added p values to the abstract as follows:

Lines 29-31: Favorable father food parenting practices were associated with adolescent intake of more fruit and vegetables and less sugar-sweetened beverages, sweets/salty snacks, and fast food (all p<0.05 or p<0.01).

Lines 32-34: Additional analyses showed favorable food parenting practices in combination with frequent family meals were associated with adolescents having higher intake of fruit (p = 0.011).

Introduction

Comment 3: Line 45 and 49 – past tense is used to describe findings and it would be better to complement with year of data collection /publication for context or use present tense …as in ‘current research shows that…’

Response 3: We did not add year of data collection because the various studies encompassed several different data collection periods. However, we revised these statements as follows to use present tense to introduce the studies (lines 47-54):

According to recent studies, Mexican-American and other Hispanic children had lower fruit and vegetable intakes [6–8], and higher sugar-sweetened beverage (SSB) [9], sweets/salty snack [10], and fast food [11] intakes than recommended by the Dietary Guidelines for Americans (DGAs) [12] and other expert groups [13, 14]. Current research shows that Hispanic children and adolescents had the highest obesity rates compared to other ethnic/racial groups, with about half of all Hispanic children and adolescents classified in overweight or obese categories based on nationally representative data [15–17].  

Comment 4: Line 52-54 reference this statement please

Response 4: We added the reference for this statement as follows (lines 56-58):

Protective factors based on Latino family strengths can interact with risk factors to address health disparities among children and adolescents in urban, low-income households [3].

Comment 5: Line 57 “several reviews and a qualitative study…” – it would be best to summarise review findings and separately discuss context using qualitative studies as do not think appropriate to aggregate here.

Response 5: We separately discussed information from review articles and the qualitative study as follows:

Lines 61-63: Several reviews have identified a variety of food and activity parenting practices that influence adolescents' food and activity behaviors, including setting expectations, role modeling, and managing availability [23, 24].

Lines 78-80: A focus group study identified eight primary food and activity parenting practices reported by Latino fathers (n = 26) related to improving their children’s healthy lifestyles [35].

Comment 6: Line 62-63 Expand this to describe (if known) any unique role of the father (as distinct from mother)

Response 6: We expanded this information to describe how father food parenting practices may differ from mothers of children 5-12 years based on a cross sectional Canadian sample of parents as follows (lines 69-75):

In a cross sectional sample of Canadian parents of children (5-12 years), relationships among food parenting practices were similar between mothers and fathers for some children’s eating behaviors, but differentially associated with behaviors regarding food and satiety responsiveness [34]. For example, paternal restriction for weight practices, practices to accommodate the child, and use of covert control were associated with higher child food responsiveness, while only maternal restriction for weight practices were associated with higher food responsiveness.

Comment 7: Line 78-79 How is it in line with social cognitive theory exactly?

Response 7: We revised the statement about Social Cognitive Theory in lines 90-93 as follows:

Environmental factors including father food parenting practices, father food/meal involvement, and family meals operate within the reciprocal determinism construct of Social Cognitive Theory to influence adolescents' dietary behaviors.

Methods

Comment 8: A RCT is mentioned in the methods, however no details are given about this. As per line 117, it seems this paper presents data from intervention and control groups at baseline before randomization? That needs to be stated more clearly.

Comment 9: We also need a bit of detail about what the intervention involves to provide context as to why they might be participating. Also personnel involved in the study, modality and location, frequency, duration etc.

Response 8 and 9: We added more details about the intervention in lines 128-137 as follows:

Baseline data were collected prior to randomization of participants into intervention and control groups. The intervention group attended 8-weekly educational sessions about nutrition, physical activity and positive food and physical activity parenting practices. The 2.5-hour educational sessions were conducted in-person at churches or community centers with trained bilingual facilitators leading the interactive sessions. Evaluation data were collected in the same settings by trained research assistants at baseline, post and 3 months after the intervention group educational sessions. The control group attended the same educational sessions after the 3-month data collection session. The data collection sessions lasted about 1 hour per father/adolescent dyad including questionnaires, height and weight measurements and adolescent dietary recall interviews.

Comment 10: What other study measures were taken given their inclusion in a wider intervention. Did you have anthropometric data for the sample? And if yes, can these be used to assess the accuracy of reported energy intake from the 24-hr recall? I think it would be important to know if the proportion of the sample defined as healthy weight/overweight/obese. The aim of the study is stated to be 'prevention of overweight and obesity' which implies that possibly they were not overweight/obese sample, but this needs clarification. It would also enhance the methods section to know in what context were the dietary behaviour data collected - alongside other questionnaires and measures?

Response 10: We collected height and weight data from the sample during data collection sessions at baseline, post and 3 months after the intervention group educational sessions. We added information to the methods section to describe how these data were collected as follows (lines 188-195):

Adolescents’ body weight and height were measured separately twice in a private space using a digital scale (BWB-800 Scale, Tanita) and a stadiometer by a trained research assistant according to standardized procedures of the National Health and Nutrition Examination Survey (NHANES) [47]. BMI percentiles were generated by a SAS program for the 2000 CDC Growth Charts and categorized as underweight (<5th percentile), normal weight (5th - <85th percentile), overweight (85th – <95th percentile), and obese (≥95th percentile) [48].

We also added results regarding adolescent BMI group to Table 1 and described the results in the results section as follows (lines 301-302):

The majority of adolescents (58%) were classified in the overweight or obese category.

We did not attempt to assess the accuracy of reported energy intake from the 24-hr recall because of the multiple confounding factors that would affect this relationship among adolescents. We added the proportion of participants by weight category to Table 1 and described the context in which the dietary data were collected in more detail in lines 169-171 as follows:

The first recall was conducted in person during the baseline data collection session, with two additional recalls completed by phone within the next 1-2 weeks.

In the Study design section, we also added the following (lines 135-137):

The data collection sessions lasted about 1 hour per father/adolescent dyad including questionnaires, height and weight measurements and adolescent dietary recall interviews.

Comment 11: What did participants know about the study from informed consent? Were all study objectives clear to participants, both fathers and children?

Response 11: We explained all procedures in the consent and assent forms for fathers and adolescents including schedules for educational and data collection sessions for intervention and control group participants. Participants consented or assented to participate based on knowledge of what they would be asked to do if they were randomly assigned to either the intervention or control group.

We added a sentence in lines 124-126 about assent and consent forms as follows:

Consent and assent forms explained all procedures involving educational and data collection sessions that fathers and adolescents were asked to complete in relation to the study objectives.

Comment 12: Line 141-142 should it be ‘analyzed’ instead of ‘completed’? I’m not sure I understand the structure of what is data collection and what is analysis in this paragraph

Response 12: We revised this statement as follows to clarify (lines 167-169):

To estimate dietary intake, 24-hour dietary recall interviews were conducted using Nutrition Data System for Research software (NDSR; Nutrition Coordinating Center, University of Minnesota).

Comment 13: Line 152 SSB – is it ‘categorized’ rather than ‘calculated’? Similarly line 154 and 157.  I’m not sure I understand, again, the difference between data collection, analysis and calculations here, it needs to be much clearer.

Response 13: We revised this statement as follows to clarify (lines 179-182):

SSB intake was calculated based on reported intake of beverages categorized by NDSR software as sugar sweetened beverages, which included sweetened soft drinks, fruit drinks, tea, coffee, coffee substitute, and water.

Comment 14: Lines160-163 I don’t think IQR needs to be explained. However, this paragraph needs to clearer – explain data entry error and also were all these adolescents’ data removed or just for certain food groups? Would removing participants showing outlier data be more appropriate for statistical consistency? Suggest integrating this with statistical analysis for consistency in reporting methods.

Response 14: We deleted explanation of IQR and moved this paragraph to the statistical analysis section. We described the removal of data from only certain food groups from 3 adolescents and removal of fruit intake data for one adolescent because of data entry error as follows (lines 258-261):

Outliers for adolescent dietary intake data were examined using histograms and the interquartile range (IQR) formula. Intake data for a particular food group were removed from three adolescents based on values above (Q3 + 1.5 x IQR) and from one adolescent based on data entry error (a researcher entered an over estimation of fruit intake).

Comment 15: Line 165 For adolescents reporting the perceived frequency of father food parenting practices - What tool was used – developed for this study? Any references that informed its development? Who administered the questions (qualification, training?), in what setting and how long did it take?

Response 15: We revised the text to explain how the survey was developed with supportive references as follows (lines 200-203):

Adolescent reported food parenting practice items and scales developed for this study were adapted from existing scales [49–51] and showed adequate criterion validity in a preliminary study and internal consistency for all scales based on Cronbach α coefficients > 0.7. [52]

We also added information about how the data were collected as follows:

Lines 132-134: Evaluation data were collected in the same settings by trained research assistants at baseline, post and 3 months after the intervention group educational sessions.

Lines 135-137: The data collection sessions lasted about 1 hour per father/adolescent dyad including questionnaires, height and weight measurements and dietary recall interviews.

Comment 16: Line 206-211 If father did not plan, buy, or prepare food with the adolescent, how was fathers involvement in meals comprehensively accounted for? For example, they planned, bought, or prepared food for the family without direct child involvement? Is that what was being asked in these questions?

Response 16: In the text (lines 243-246) we included the questions that were used to assess this construct as follows:

Father food/meal involvement was examined by asking adolescents three questions: "How often does your father plan meals together with you?" "How often does your father buy foods together with you?" and "How often does your father prepare foods together with you?

The questions were intended to assess only the frequency of father meal/food involvement with direct child involvement. These questions did not measure what fathers did without direct child involvement and therefore did not represent a comprehensive assessment of fathers' involvement in food/meals.

Results

Comment 17: Line 249 What is GED?

Response 17: We revised the text in lines 290-294 as follows:

The distribution of father educational attainment showed that 20% had completed some college or more, 43% had a high school diploma or GED (General Education Development test that shows high school academic knowledge), and 37% had not completed high school.

Comment 18: Line 247 would it be clearer to say ‘living with a spouse or partner?’

Response 18: We revised the text as follows (lines 289-290).

Slightly more than half of the fathers were 41 years or older (56%), with 92% living with a spouse or partner.

Comment 19: Line 251 37% or over one third being food insecure seems like a more notable finding and perhaps worth focusing on their characteristics here.

Response 19: We revised the text as follows to note the percentage being food insecure (lines 295-296):

The majority reported being food secure (63%) while 37% reported being food insecure.

Comment 20: Line 253-255 Phrasing confusing here, ‘vs. less often’ - state more clearly.

Response 20: We revised the text as follows (lines 298-301):

Nearly half of the adolescents (47%) reported having family meals ≥ 7 times a week vs. ≤ 6 times a week (Table 1). About half (49%) reported that their father was involved often or always vs. never to sometimes in planning meals, buying, and preparing foods with them.

Discussion

Comment 21: Lines 372-377 what was known about father involvement in food planning, shopping, and preparation among your sample, as reported by fathers? Good to include this alongside discussion here and if not known, important to acknowledge as limitation. Anything known about mothers or partners that could be shared in the findings for this study or integrated into the discussion.

Response 21: We added information about father food/meal involvement based on adolescent responses to this discussion (lines 419-420) as follows:

In the current study, about half of the adolescents reported that fathers were often or always involved in planning, buying, and preparing foods with them.

Adolescents only reported on father food/meal involvement, therefore we could not share adolescents’ perceptions of mother food/meal involvement. We added this as a limitation in lines 501-503 as follows:

Finally, data on adolescent perceptions of mothers’ parenting practices and food/meal involvement were not collected, therefore these findings could not be shared or integrated into the discussion.

Comment 22: Line 378-386 the sample within the study were different to general Latino community as they were already enrolled on a father/child health intervention – how does that influence this aspect of the discussion?

Response 22: We added the following at the end of the paragraph discussing father limits on fast food intake in lines 430-433 to address this comment:

However, youth in the current study were interested in enrolling in a father/adolescent nutrition and physical activity health intervention, and therefore may not have had similar perceptions regarding their fathers’ limits on fast food intake as a general Latino adolescent population.

Comment 23: Line 418-420 Is there anything known about adult male Latino snacking (similar to child data) from national data?

Response 23: We were only able to find a reference for nationally representative snacking data for Hispanic adults, which were not stratified by sex. We added information about this study to the discussion in lines 466-468:

Nationally representative data between 1977 and 2012 also showed that average energy intake from snacks per Hispanic adult significantly increased from 167 to 418 kcals per day [66]

Comment 24: Line 421-423 Rephrase this sentence to ensure relevance of latter element of the statement is clear

Response 24: We revised the sentence as follows (lines 462-471):

According to the findings from dietary recalls of eight nationally representative surveys from 1977 to 2014, average energy intake from snacks among Mexican-American children aged 2-18 significantly increased from 205 to 453 kcals per snacking occasion [10]. Based on adolescent perceptions from the current study, Latino fathers' engagement in parenting practices that lower adolescent sweets/salty snack intake may be an important intervention target to decrease intake of energy-dense, nutrient-poor foods consumed as snacks.

Limitations

Comment 25: It is a very decent sample size, however a ‘large population-based sample’ needs further explanation as in the methods it is a convenience sample, therefore likely biased in certain directions (as also alluded to). It would be good to include more context about recruitment and what kind of selection biases were noted for the sample.

Response 25: We revised the sentence about selection bias adding more details about sites:

Revised sentence is in lines 497-500:

Furthermore, two study sites were community centers which regularly offered other health and nutrition classes, thus the fathers and youth who agreed to participate in the study might have enrolled due to their interest in nutrition and health, which makes them different from the general population.

Comment 26: The (possible) influence of the monetary incentive also needs discussion in the limitations. Many potential confounding factors are not acknowledged in the limitations for this study and need to be considered, what is not known about the sample? – e.g. information about maternal/partner practices and how these explain or influence family food behaviours (for both father and child). Have any studies looked at maternal and paternal practices together and if yes, what can these types of studies tell us?

Response 26: We added a sentence about monetary incentives and adolescent perception of mother practices in the limitation part as follows

Line 500-501: Monetary incentives may also have influenced decisions to enroll in the study.

Lines 501-503: Finally, data on adolescent perceptions of mothers’ parenting practices and food/meal involvement were not collected, therefore these findings could not be shared or integrated into the discussion.

Round 2

Reviewer 3 Report

Thank you for your responses and revised manuscript.  There are two new pieces of information provided by you on revision that need further clarification or integration:

  1. Overweight and obesity in sample provided in Table 1 - what was the eligibility criteria?  58% were overweight or obese, and it is still stated in abstract and introduction that 'obesity prevention' was the aim of the programme.  This needs to be consistent throughout.
  2. lines 203-206:  "However, father-reported parenting practice items showed limited criterion validity, and significant discrepancies were identified between adolescents’ and fathers’ reports of parenting practices. Therefore, adolescent reported food parenting practices data were used in this study."  This seems to present difficulty in the accuracy of your data and warrants further discussion about which party provides most accurate data.  I appreciate you have changed the title and pivoted the article, however you do not explain this observation or show us how they report differently, which I think is an important elements of your findings.

Author Response

Hi,
Thank you very much for your feedback. We made all of the changes based on your feedback on the paper. Also provided the point by point responses to your comments below:

  1. Overweight and obesity in the sample provided in Table 1 - what was the eligibility criteria?  58% were overweight or obese, and it is still stated in the abstract and introduction that 'obesity prevention' was the aim of the programme. This needs to be consistent throughout.

The intervention was intended to prevent overweight among those categorized as normal weight, to prevent obesity among those categorized as overweight, and to prevent severe obesity among those categorized as obese. In our sample, 41% were categorized as normal weight, 26% were categorized as overweight, and 32% were categorized as obese. We added the following to lines 143-147:

Eligibility criteria for adolescents included being the child of a Latino father/caregiver and being 10-14 years of age. The intervention was intended to prevent overweight among those who were categorized as normal weight at baseline, prevent obesity among those categorized as overweight, and prevent severe obesity among those categorized as obese.

For consistency throughout, we revised the paper as follows:

Abstract - Lines 23-24: Baseline data were used from Latino adolescents (10-14 years, n = 191, 49% boys) participating with their fathers in a community-based overweight/obesity prevention intervention.

Introduction - Lines 59-61: Both maternal and paternal caregivers have an important role in preventing childhood overweight and obesity through the formation of healthy food-and activity-related behaviors among youth [21, 22].

Methods - Lines 118-121 were not changed as they already focused on prevention of overweight and obesity as follows: The randomized controlled intervention trial (Identifier: NCT03641521) aimed to prevent overweight and obesity among Latino adolescents (10-14 years) by improving father food- and physical activity-parenting practices and youth energy balance-related behaviors.

Conclusions - Lines 524-526: These findings emphasize the importance of including fathers in behavioral and family-based research to prevent overweight and obesity among adolescents.

  1. lines 203-206:  "However, father-reported parenting practice items showed limited criterion validity, and significant discrepancies were identified between adolescents’ and fathers’ reports of parenting practices. Therefore, adolescent reported food parenting practices data were used in this study."  This seems to present difficulty in the accuracy of your data and warrants a further discussion about which party provides the most accurate data. I appreciate you have changed the title and pivoted the article, however, you do not explain this observation or show us how they report differently, which I think is an important element of your findings.

We expanded the text in the Methods section regarding criterion validity for adolescent- vs. father-reported frequency of parenting practices to further explain this observation and rationale for using adolescent vs. father report (lines 204-221).

Adolescent-reported food parenting practice items and scales developed for this study were adapted from existing scales [50–52] and showed internal consistency for all scales based on Cronbach α coefficients > 0.7 in a preliminary study [53]. Adequate criterion validity was demonstrated for 19 of the 21 parenting practice measures based on the adolescent report. Criterion validity was indicated by significantly higher adolescent-reported consumption of fruit and vegetables; lower consumption of SSBs, sweets/salty snacks, and fast foods; greater weekly physical activity hours; and fewer daily screen time hours among adolescents who reported high vs. low levels/frequencies of supportive parenting practices. However, father-reported parenting practice items and scales only showed criterion validity for 3 of the 21 parenting practice measures. These results indicated greater consistency between perceived frequency of paternal parenting practices and adolescent behaviors when adolescents reported parenting practice frequency vs. fathers. The percentage agreement between adolescent- and father-reported dichotomized responses regarding paternal parenting practice frequency varied from 49% to 68% for expectations/limits, 51% to 70% for modeling, and 52% to 70% for availability practices. In general, adolescents reported lower frequencies of supportive food parenting practices than fathers. Adolescent-reported food parenting practices data were used in this study instead of father-reported data based on these preliminary testing results [53].